# Application of Ionic Liquids for Chemical Demulsification: A Review

**DOI:** 10.3390/molecules25214915

**Published:** 2020-10-23

**Authors:** Nahid Hassanshahi, Guangji Hu, Jianbing Li

**Affiliations:** 1Environmental Engineering Program, University of Northern British Columbia, Prince George, BC V2N 4Z9, Canada; Hassansha@unbc.ca; 2School of Engineering, University of British Columbia, Kelowna, BC V1V 1V7, Canada

**Keywords:** ionic liquids, emulsion, chemical demulsification, interfacial tension

## Abstract

In recent years, ionic liquids have received increasing interests as an effective demulsifier due to their characteristics of non-flammability, thermal stability, recyclability, and low vapor pressure. In this study, emulsion formation and types, chemical demulsification system, the application of ionic liquids as a chemical demulsifier, and key factors affecting their performance were comprehensively reviewed. Future challenges and opportunities of ionic liquids application for chemical demulsification were also discussed. The review indicted that the demulsification performance was affected by the type, molecular weight, and concentration of ionic liquids. Moreover, other factors, including the salinity of aqueous phase, temperature, and oil types, could affect the demulsification process. It can be concluded that ionic liquids can be used as a suitable substitute for commercial demulsifiers, but future efforts should be required to develop non-toxic and less expensive ionic liquids with low viscosity, and the demulsification efficiency could be improved through the application of ionic liquids with other methods such as organic solvents.

## 1. Introduction

The presence of emulsion in oil or in water has undesired consequences for industries and the environment [1]. It may result in the corrosion of pumps, pipes, and related facilities. It increases the viscosity of oil, which leads to the increment of pumping and transporting costs, and the emulsion also reduces the quality of oil [2]. Discharging produced oily wastewater from oil and gas industries and the spill of oil through ship accidents and offshore wells (e.g., Exxon Valdez and Deepwater Horizon) [3,4] into waters cause adverse consequences to the environment, human health, and the economy [5,6,7]. The oil spill cleanup usually involves the collection of a large volume of oily wastewater for treatment. Strict limitations are regulated for discharging oily wastewater (e.g., based on USEPA, oil and grease discharge limits are 29 mg/L monthly average and 42 mg/L daily maximum) which induce industries to efficiently treat their oily wastewater [8]. However, the presence of emulsion in oily wastewater essentially requires a demulsification for its effective treatment. There are several technologies that have been used for separating oil and water, such as various physical (e.g., gravitational settling, thermal treatment, membrane separation, flotation, ultrasonic), biological (bioaugmentation, biostimulation), and chemical (e.g., solidifiers, demulsifiers, sorbents) processes [9,10,11,12,13,14,15,16,17]. Chemical demulsification using various demulsifiers is one of the reliable methods which has been widely used in industries for breaking emulsions [15,18].

Many research studies have been conducted for the application of ionic liquids in chemical demulsification processes, and it was reported that ionic liquids are reliable demulsifiers with high stability, even under harsh conditions (high temperature and high salinity) [19]. Ionic liquids are produced by a combination of different organic cations and organic or inorganic anions [20]. They are associated with unique characteristics such as thermal stability, non-flammability, recyclability, and low vapor pressure [21,22,23]. These properties make ionic liquids a suitable substitute for organic solvents and commercial demulsifiers [23,24]. Ionic liquids have been used by different researchers to evaluate their efficiency in demulsification processes, mostly at the laboratory scales. There is a need for a comprehensive discussion of current ionic liquids demulsification method to identify its advantages and limitations. In this study, a review on the application of ionic liquids for demulsification was conducted, and the impacting parameters on the demulsification performance were discussed to identify challenges and opportunities for future applications. This review is organized into a few sections, including emulsion formation mechanisms and types, chemical demulsification system and ionic liquids application, the effects of influential parameters (concentration, cation type, and the structure of ionic liquids, anion types, molecular weight, salinity, temperature, and oil types), as well as challenges and opportunities for future applications.

## 2. Emulsions

### 2.1. Emulsion Formation

Emulsion is produced when two or more immiscible liquids mix vigorously together which results in two phases (a dispersed phase and a continuous phase) [25,26]. The phase that has smaller volume is usually identified as the dispersed phase and the larger one is the continuous phase. If the volume of both phases are the same, other factors would be considered to recognize the dispersed and continuous phases [27,28]. Based on the Bancroft rule, a continuous phase would be the phase that emulsifying agents are more soluble in it [29]. Commonly investigated emulsions have a water phase and an oil phase. Fine solids and surface active compounds of crude oils, such as saturates, asphaltenes, resins, and aromatics, can act as natural emulsifying agents [30,31]. Under intensive mixing, natural emulsifying agents could adsorb at the oil–water (O-W) interface, creating a rigid interfacial film around dispersed droplets and hinder the coalescence of droplets [32,33,34]. The strong tension between water and oil phases is called interfacial tension (IFT), and the stronger the IFT, the more stable the emulsion [35].

### 2.2. Emulsion Types

According to the nature of the dispersed phase, emulsions are categorized into oil in water (O/W), water in oil (W/O) and multiple (W/O/W or O/W/O) types (Figure 1). O/W emulsions occur when oil droplets are the dispersed phase (inner phase) in the continuous water phase (outer phase), which is also called reverse emulsion. W/O emulsions are generated when water droplets are the dispersed phase in the continuous oil phase. Multiple emulsion is a mixture of W/O and O/W emulsions. O/W/O emulsions are formed when oil droplets are the dispersed phase in water droplets that are dispersed in the continuous oil phase, while W/O/W emulsions are formed vice versa. The occurrence of multiple emulsions is more common in food, cosmetics, pharmaceutics, and wastewater treatment industries [36,37,38,39]. It should be noted that these emulsions are thermodynamically unstable but are kinetically stable. Emulsions are thermodynamically unstable because they are produced from the mixture of two or more immiscible liquids which naturally tend to separate. However, the kinetic stability of emulsions means that emulsions are stable for weeks to years due to the formation of strict films around them by emulsifiers [40,41].

The aforementioned emulsions can be categorized into stable, mesostable, entrained water, and unstable, depending on the time that one phase can be kept dispersed in the other phase [15,30,42]. In general, stable W/O emulsions can hold most of the water in the oil phase for more than five days [43]. Mesostable emulsions are stable within only one to three days, while entrained water and unstable W/O emulsions are not regarded as stable emulsions because both would only remain water in oil for less than one day [26,42]. Emulsion type is an important factor in selecting demulsifiers which are soluble in the continuous phase and could reach the O-W interface easily [36]. Emulsion type depends on the affinity of natural emulsifying agents to the oil (or water) phase. If natural emulsifying agents have tendency to the oil phase (i.e., hydrophobic natural emulsifying agents), W/O emulsions would form, while the hydrophilic natural emulsifying agents would produce O/W emulsions. The same propensity of natural emulsifying agents to both oil and water phases would lead to the formation of unstable emulsions. The factors determining the affinity of natural emulsifying agents include hydrophilic-lipophilic balance (HLB), hydrophilic–lipophilic deviation (HLD), relative solubility number (RSN), and R ratio [44,45,46,47]. Highly affinity of natural emulsifying agents to a lipophilic or a hydrophilic phase leads to the formation of less stable emulsions because natural emulsifying agents tend to stay in a medium rather than migrate to the O-W interface [48].

## 3. Ionic Liquid Demulsification

### 3.1. Chemical Demulsification System

Surface-active chemicals (i.e., demulsifiers) are used to destabilize emulsions [25,49]. Some examples of common chemical demulsifiers include sodium dioctyl sulfosuccinate, sodium dodecyl sulfate, and polyethylene oxide (Figure 2). The surface activity of demulsifiers should be higher than that of natural emulsifying agents to destabilize the emulsion [36]. Surface activity features of demulsifiers can be evaluated by surface tension, electrical conductance, fluorescence, proton nuclear magnetic resonance (H NMR) and small angle neutron scattering (SANS) methods [50]. Chemical demulsification is implemented by adding a desired amount of demulsifier to emulsions and mixing them vigorously. After mixing, sufficient time is required to allow Ostwald ripening, flocculation, coalescence, and phase separation (creaming/sedimentation) to occur. Ostwald ripening occurs when a dispersed phase can diffuse easily in a continuous phase to reach together for coalescence. Flocculation is formed when oil or water droplets flock together in a continuous phase while they keep their identity. Coalescence is an irreversible process which water or oil droplets join together and make bigger droplets. Creaming or sedimentation processes occur based on the density of a dispersed phase [31,51]. The general chemical demulsification mechanism is shown in Figure 3.

### 3.2. Applicaion of Ionic Liquids and Their Characteristics

Ionic liquids were first introduced by Paul Walden in 1914 when he discovered a special chemical-ethyl ammonium nitrate ([EtNH_3_] [NO_3_]) with a melting point of 12 °C [52]. Ionic liquids have been widely used in different fields like pharmaceutical, oil and gas, and chemical industries [53,54,55,56,57,58], such as in pharmaceutical products manufacturing [59,60,61,62], viscosity modifiers [63], desulfurization of liquid fuels [64], and liquid-liquid extraction [65,66,67]. Several types of ionic liquids, such as polymeric ionic liquids [68,69,70], double salt ionic liquids [71], dicationic ionic liquids [72,73,74,75], deep eutectic solvents [76,77,78,79], chiral ionic liquids [80], and solvate ionic liquids [81,82], have been synthesized for different aforementioned purposes.

Around ten^18^ ionic liquids can be synthesized by combining organic cations and organic or inorganic anions which are non-flammable liquid salts with melting point below 100 °C [21,83]. Ionic liquids with melting point below room temperature (∼25 °C) are called room temperature ionic liquids (RTIL) [84]. Table 1 lists some common ionic liquids with their melting points [85]. Ionic liquids have lower vapor pressure than conventional volatile organic solvents, and thus present less risk to the environment. They have been applied in many industries as a replacement of conventional volatile organic solvents and significantly reduced the generation volume of hazardous wastes [86,87]. Ionic liquids are used successfully in enhanced oil recovery (EOR) processes. Pillai et al. [88] investigated the effect of C_8_mim BF_4_, C_10_mim BF_4_, and C_12_mim BF_4_ on EOR and IFT reduction of O-W solution at the temperature of 30 °C. The results indicated that C_8_mim BF_4_, C_10_mim BF_4_ and C_12_mim BF_4_ reduced IFT to 14.57 mN/m, 4 mN/m, and 2.1 mN/m at the concentration of 12,000 ppm, 5000 ppm, and 2000 ppm, respectively. C_12_mim BF_4_ was applied for EOR which recovered 32.28% of oil additionally in ionic liquid (1.2 × concentration (2000 ppm), 0.5 pore volume), PHPA polymer (2000 ppm, 0.5 pore volume), and organic alkali triethylamine (1%, 0.5 pore volume) flooding setup [88].

The properties of ionic liquids are essential for the health and safety concerns in industries. Ionic liquids can remain structural stable even at high temperatures (e.g., 300 °C), but other conventional surfactants may degrade at those temperatures. Such characteristics make ionic liquids to be unique compared to other demulsifiers [86]. Most researchers prefer to design and synthesize RTIL. RTIL can be achieved by synthesizing nitrogen or phosphorous organic cations with different organic anions, dicyanamide, acetate, trifluoromethylsulfate or inorganic anions, bromide, chloride, tetrafluoroborate, hexafluorophosphate [84]. Figure 4 shows a few common cations of ionic liquids, including imidazolium, pyridinium, pyrrolidinium, ammonium, and phosphonium. Table 2 lists some anions that are used for synthesizing ionic liquids.

Some ionic liquids possess amphiphilic structure which enables them to have affinity to both water and oil phases [15,18,20]. The amphiphilic character may be in cation or anion part of ionic liquids structure. Depending on the location of amphiphilic structure, ionic liquids can be classified into cationic or anionic ionic liquids [15]. Figure 5 shows the common structure of ionic liquids and the structure of 1-butyl-3-methylimidazolium chloride.

The characteristics of ionic liquids such as melting point, thermal stability, and viscosity can be modified by using different combinations of cations and anions to achieve different desirable purposes [92,93]. The water solubility (i.e., hydrophobicity and hydrophilicity), viscosity, and melting points of ionic liquids are dependent on the type, size, and structure of the anions, respectively. Bis(trifluoromethanesulfonyl)amide, bis(pentafluoroethanesulfonyl)amide, and tris(trifluoromethanesulfonyl)methanide are examples of anions that can be used to produce hydrophobic ionic liquids in combination of cations, such as 1,3-dialkylimidazolium and N-alkylpyridinium, tetraalkylammonium. Tosylate, trifluoroacetate, and dicyanamide are common anions that can be used to produce hydrophilic ionic liquids [93].

The hydrophobicity (or hydrophilicity) of anions affects the thermal stability of ionic liquids: the more hydrophilic the anions, the less thermally stable the ionic liquid [90,94,95]. Ionic liquids with a small anion size tend to have a low viscosity due to the low tendency to participate in hydrogen bonding as well as the diffusive negative charge [93]. The symmetry of anion and cation of ionic liquids affect their melting points: the less symmetry of anions and cations, the lower melting point of ionic liquids [90,96]. Increasing the cation alkyl chain length (e.g., 3–5 carbon atoms) could result in a decreased melting point of ionic liquids [93]. Nevertheless, increasing the alkyl chain length of ionic liquids could increase the thermal stability, hydrophobicity, and surface active area of ionic liquids [34,55,97].

### 3.3. Demulsification Mechanism of Ionic Liquids

The demulsification mechanism of ionic liquids involves two main steps, including diffusion and adsorption. The diffusion process is the distribution of ionic liquid molecules in the continuous phase before arriving at the O-W interface, while the adsorption process means that the diffused ionic liquid molecules pass through the continuous phase and reach the O-W interface [92,98,99,100]. The ionic liquid molecules then substitute natural emulsifying agents at the interface and change the viscoelastic properties of the interfacial films. This leads to breaking the strong film around O-W droplets and enhancing the coalescence of the dispersed droplets [15,33,34].

Recent investigations have found that hydrophobic surface-active ionic liquids can be used for effective demulsification of W/O emulsions in the oil and gas industries [24,93]. A research conducted by Hazrati et al. [101] indicated that hydrophobic ionic liquids (e.g., C_n_mim PF_6_) demulsified emulsions better than hydrophilic ionic liquids (e.g., C_n_mim Cl) [101]. To facilitate the dissolution of ionic liquids in oil phase, organic solvents such as xylene and methanol can be used along with hydrophobic and hydrophilic ionic liquids, respectively [36]. Dichloromethane, chloroform, isopropanol, ethanol, benzene, and toluene can also be used individually or in mixtures to achieve the same purpose [102]. Tian et al. [103] used C_2_mim BF_4_ with cyclohexane for enhancing oil recovery from tank bottom oily sludge (a stable emulsion), and they found that more than 95% of total petroleum hydrocarbons recovery can be achieved at 0.1 mL/g of ionic liquid/sludge ratio, a time of 10 min, solvent/sludge of 4:5 mL/g, and a shaking speed of 100 rpm [103]. Table 3 categorizes the common solvents which are used for synthesizing as well as for facilitating the dissolution of ionic liquids in oil phase into three groups, including preferred, usable, and undesirable, based on their physical and chemical properties, toxicity, environmental and safety aspects, operational concerns, and costs [104,105].

To evaluate the demulsification efficiency of ionic liquids, bottle test is commonly used in laboratories [15,18,34]. In a bottle test, an ionic liquid is added dropwise to a graduated settling tube containing emulsions. Then their mixture is shaken for 1–5 min, followed by the measurement of the height of the separated water and oil at different time [18]. The demulsification efficiency is calculated using Equation (1) [14]:(1)$$\mathrm{DE}=\frac{C_{i}-C_{f}}{C_{i}}\times 100$$
where DE is demulsification efficiency, C_i_ is the initial oil (water) concentration of emulsion, and C_f_ is the final oil (water) concentration of emulsion.

## 4. Factors Affecting Ionic Liquids Demulsification

Many researchers applied different types of ionic liquids to investigate their effects on demulsification processes. A number of factors were found to affect the demulsification performance of ionic liquids, and they are described below.

### 4.1. Concentration

The concentration of ionic liquids could affect the demulsification efficiency. Generally, increasing ionic liquid concentration up to reaching micellization would increase the demulsification efficiency. When water at the O-W interface saturates with the hydrophilic parts of an ionic liquid, micellization happens. The concentration of ionic liquids that can initiate micellization is called critical micelle concentration (CMC). The CMC of ionic liquids is identified by measuring the IFT of a solution at different concentrations of ionic liquids: when the IFT is the minimum, the concentration is identified as the CMC [106]. However, using ionic liquids at concentrations higher than CMC will not lead to any significant change in IFT [100], and a demulsifier concentration higher than the CMC could lead to adverse effects on a demulsification process because ionic liquid molecules aggregate and become an emulsifier agent [106,107]. Moreover, the relationship between IFT reduction and demulsification efficiency is still not well understood [18]. Bin-Dahbag et al. [106] investigated the effect of tetraalkylammonium sulfate and its concentration (100–1000 ppm) on the efficiency of IFT reduction in saline W/O emulsions (10% and 20% w/w salinity), and they observed that by increasing ionic liquid concentration to CMC (250 ppm), the IFT reduced from 18 to 3.36 mN/m and from 14.5 to 1.65 mN/m for 10% and 20% salinity of solutions, respectively, while increasing ionic liquids concentration above CMC had no significant changes on IFT reduction [106]. In a research conducted by Hezave et al. [98], the effect of C_12_mim Cl on IFT reduction in O/W emulsion at different ionic liquid concentration (0–5000 ppm) and water salinity (10,000–100,000 ppm) was investigated, and they found that by increasing concentration of ionic liquid to CMC (100 ppm), IFT reduced noticeably (from 38.02 to 0.81 mN/m), but no significant change in IFT was observed by increasing the concentration of ionic liquids beyond CMC [98].

### 4.2. Cation Type and Structure of Ionic Liquids

In addition to the applied concentration, the effectiveness of ionic liquids in demulsification processes depends on the cation type and cation alkyl chain length [18,50]. Ionic liquids and their cations must have a high molecular volume (e.g., 1000–1500 Å^3^) to function effectively as a demulsifier. Larger cation volume (e.g., 900–1400 Å^3^) increases the polarizability of the cation which leads to higher demulsification efficiency [108]. Sakthivel et al. [109] used lactam and imidazolium based ionic liquids for EOR and IFT reduction, and they evaluated the effectiveness of ionic liquids at the concentration of 5000 ppm under zero and high salinity (100,000 ppm) conditions [109]. They observed that both ionic liquids performed better at high salinity than zero salinity condition because of the interaction of ionic liquids and salt ions at the O-W interface, and they demonstrated that the lactam-based ionic liquid was better than the imidazolium-based ionic liquid in IFT reduction and EOR because the former one had more polar moieties in the structure [109].

Cation alkyl chain length and its structure (e.g., straight or branched) could change the properties of ionic liquids. Molecular polar-polar interactions between the polar fractions of oil and the polar moieties of ionic liquid would increase by increasing the cation alkyl chain length [109]. Saien et al. [110,111,112] indicated that increasing alkyl chain length of C_n_mim Cl ionic liquid from 6 to 16 significantly decreased the IFT of n-butyl acetate-water at 25 °C [110,111,112]. Ionic liquids with branched and long alkyl chain are more hydrophobic and have lower CMC than those with short and straight alkyl chain [34,92,99]. However, it should be noted that very long alkyl chain might impede ionic liquids to reach the O-W interface of W/O emulsions. Guzman-Lucero et al. [113] synthesized ionic liquids with different alkyl chain length (5–18 carbon atoms) to demulsify W/O emulsions at the ionic liquid concentration of 1000 ppm and the temperature of 80 °C, and they indicated that the ionic liquid with 18 carbon atoms had lower efficiency in demulsification of W/O emulsions than ionic liquids with 12 and 14 carbon atoms [113]. The surface activity of ionic liquids varies based on their cation types and cation alkyl chain length. Sastry et al. [50] evaluated the surface activity of ionic liquids with different cation type (methylimidazolium, methylpiperidine, methylpyrrolidine) and alkyl chain length (10, 12, 14, 16, 18 carbon atoms) by measuring surface tension and solution conductivity, and they observed that the surface activity of ionic liquids increased by increasing the alkyl chain length. The surface activity of methylimidazolium-based ionic liquids was higher than methylpiperidine and methylpyrrolidine-based ionic liquids with the same carbon alkyl chain (octadecyl) and anion type (chloride) [50].

### 4.3. Anion Type of Ionic Liquids

Investigations indicated that hydrophobicity and hydrophilicity of anions as well as the size of anions are important factors influencing demulsification efficiency [101,114]. Hydrophobic ionic liquids with a large anion size, such as C_8_mim PF_6_, can reduce the chance of aggregation formation of ionic liquid molecules in a solution, which leads to an enhanced demulsification process and IFT reduction [114]. These ionic liquids can obtain a high demulsification efficiency even at low concentrations [92,101,115]. In other words, anions that have weaker hydration (i.e., high polarizability) would adsorb greatly at the O-W interface, break the strict films around droplets, and enhance the demulsification efficiency. For example, ionic liquids containing bromide anions in their structure performed better than those with chloride anions because bromide has weaker hydration than chloride [50]. Saien et al. [110] compared different halide anions (I, Br, Cl)^−^ based ionic liquids and found that ionic liquids with bigger anion size (I^−^ > Br^−^ > Cl^−^) were more polarizable and adsorbed better at the O-W interface. Based on the results, C_16_mim I, C_16_mim Br, and C_16_mim Cl reduced IFT from 13.4–14.0 mN/m to 3.7, 3.9, and 4.0 mN/m at the concentration of 2.5 × 10^−3^ mole/L and the temperature of 25 °C, respectively [110]. Abdullah et al. [116] investigated the effect of anion type of GEB-Cl and GEB-TFA ionic liquids on the demulsification efficiency and IFT reduction of sea W/O emulsions (oil:seawater of 50:50, 70:30, 90:10 volume %) at different ionic liquids concentration (250, 500, 1000 ppm), and they found that increasing concentration to 1000 ppm increased the demulsification efficiency (100%) and reduced the IFT of emulsions from 33.5 mN/m to 8.4 mN/m and 7.2 mN/m for GEB-Cl and GEB-TFA, respectively. Based on their result, the greater hydrophobicity of trifluoroacetate anion compared to chloride anion led to better IFT reduction by GEB-TFA than GEB-Cl [116]. Some of the recent studies are summarized in Table 4.

### 4.4. Molecular Weight

The molecular weight of ionic liquids could affect their molecules movability and diffusion through a continuous phase [34,87]. Demulsifiers with a molecular weight > 10,000 Daltons (Da) are known as high molecular weight demulsifiers. These demulsifiers have low diffusion ability and require relatively long time to function [119]. However, high molecular weight demulsifiers were reported capable of flocculating small water droplets in the continuous oil phase and destabilizing them [119,120]. Low molecular weight demulsifiers (i.e., <3000 Da) diffuse quickly in a continuous phase, and they possess high interfacial activity which can easily absorb onto the O-W interface and weaken the rigid films around droplets [34,119,120]. However, a high dosage of low molecular weight demulsifiers may be required for successful demulsification. Wu et al. [121] evaluated the effect of 52 nonionic demulsifiers on demulsification of W/O emulsions at the temperature of 80 °C, and they observed that commercial demulsifiers from four families (Span, Brij, Tween, and Igepol) were ineffective at low applied concentrations (300–400 ppm). They concluded that using demulsifiers with a molecular weight between 7500 to 15,000 Da could lead to a higher demulsification efficiency than using a demulsifier with a molecular weight of 4000 Da [121]. Commonly used demulsifying ionic liquids usually have a molecular weight < 1000 Da [19,49,115,122]. Balsamo et al. [19] in their research investigated the effect of TOMAC and C_8_mim PF_6_ ionic liquids on the demulsification efficiency at different concentrations (2.5 × 10^‒3^, 1.2 × 10^‒2^ and 2.9 × 10^‒2^ mole/L), and their results indicated that TOMAC separated the water from the oil effectively (74%) at the concentration of 2.9 × 10^‒2^ mole/L because it was more hydrophobic and had higher molecular weight (404 Da) than C_8_mim PF_6_ (340 Da) [19].

### 4.5. Salinity

The presence of salt in the water phase can help to improve the demulsification performance of ionic liquids by two means. First, salt anions in the solution (e.g., Cl^−^) can reduce the electrical repulsions between positive homonymous charges of ionic liquids at the O-W interface. This enables ionic liquids to saturate the interface completely, leading to the reduction of IFT and thus enhancement of the demulsification process [24,98,99,100]. Second, the cations of salts (e.g., Na^+^) have a smaller molecule size and a higher surface charge density than the cations of ionic liquids. Therefore, cations of salts tend to adsorb the water and induce the ionic liquids molecules to accumulate at the O-W interface [92]. This phenomenon is known as salting out which enhances the demulsification process and IFT reduction [48]. For example, Bin-Dahbag et al. [106] found that salinity (10% w/w) contributed to the reduction of IFT noticeably by improving the distribution of ionic liquid molecules at the oil-brine interfaces [106].

Salinity has more significant effects on imidazolium-based ionic liquids than on pyridinium-based ionic liquids. Imidazolium cation is more hydrophilic than pyridinium cation, which leads to better adsorption of imidazolium-based ionic liquids onto the water film at the O-W interface. The presence of salt anions in the water reduces the repulsions among imidazolium cations and results in better saturation of ionic liquids at the interface. However, as pyridinium cations are more hydrophobic than imidazolium cations, they tend to immerse to the oil phase where the anions of water have less effect on them. To confirm this, Hezave et al. [98,99] used pyridinium and imidazolium based ionic liquids to investigate their efficiency on IFT reduction of W/O emulsions with and without salt ions in the water, and they found that the IFT and CMC of ionic liquids for emulsions containing salt ions (~100,000 ppm) reduced noticeably, while no research has indicated that conventional surfactants are effective to reduce IFT at high salinity [98,99]. They also indicated that the CMC of C_12_mim Cl reduced from 2000 ppm to 100 ppm in the presence of salt ions, while that of C_12_Py Cl reduced from 500 ppm to 250 ppm [99]. Sakthivel et al. [109] investigated the effect of salinity on reducing IFT of O-W using CP C_6_H_13_COO ionic liquid, and they observed that the IFT of solution reduced from 39 to 15 mN/m and 10 mN/m at distilled water and saline water (100,000 ppm) conditions, respectively. They also concluded that the selected ionic liquid was stable and effective in high salinity compared to the other conventional surfactants such as sodium dodecyl sulfate [109].

Salinity would also affect the duration of demulsification process when using ionic liquids. Borges et al. [48] found that salinity influenced the demulsification process by reducing the O-W separation time. Adewunmi et al. [49] indicated that, when using three types of ionic liquids (trihexyltetradecylphosphonium chloride, trihexyltetradecylphosphonium decanoate, trihexyltetradecylphosphonium dicyanamide) at 80 °C, the demulsification time of W/O emulsions reduced from 10 to 5 min when saline W/O substituted distilled W/O [49]. In contrast, Lemos et al. [114] observed that increasing the salinity of aqueous phase from 0 to 50,000 ppm reduced the demulsification efficiency of C_8_mim PF_6_ from 54.7% to 27.1%, but no salinity-caused effect was observed for the demulsification process with the use of C_8_mim BF_4_ [114].

### 4.6. Temperature

Temperature can affect the physical properties of emulsion such as viscosity. The viscosity of a continuous phase reduces at high temperatures (e.g., 70 °C) [98], which facilitates the dissolution of ionic liquids in the continuous phase [25,27,107]. Temperature should be increased up to the phase inversion temperature (PIT), at which emulsion alteration occurs (e.g., W/O turns into O/W) [123]. A temperature higher than PIT enables the saturation of O-W interface by ionic liquid molecules, and facilitates the distribution of ionic liquid molecules in the continuous phase [92,100]. Hezave et al. [98,100] found that by increasing the temperature of W/O emulsion higher than the PIT (e.g., 20 °C in their research), the IFT of the emulsion increased due to the distribution of C_12_mim Cl in the continuous oil phase [98,100]. However, Bin-Dahbag et al. [106] indicated that increasing the temperature from 22 °C to 90 °C had negligible effect on the IFT reduction using tetraalkylammonium sulfate as a demulsifier [106].

Increasing the temperature of emulsion can significantly reduce its viscosity. However, it is difficult to differentiate the demulsification enhancement effects brought by reduced viscosity and by ionic liquids [19]. Balsamo et al. [19] investigated the effect of temperature (30, 45, and 60 °C) on the demulsification process using TOMAC and C_8_mim PF_6_, and they concluded that by increasing temperature from 30 °C to 45 °C, the demulsification efficiency increased for samples containing the two ionic liquids. However, increasing temperature to 60 °C resulted in a high demulsification efficiency for all the samples (with and without ionic liquids), and the reason was that great reduction in the viscosity of oil phase occurred at the temperature of 60 °C, which facilitated the coalescence of water droplets for increased settling [19].

### 4.7. Oil Types

Since the physiochemical properties of oil (e.g., density, viscosity, and natural emulsifying agents) vary greatly from field to field [33,124], the demulsification performance of the same ionic liquid on emulsions containing different types of oil might not be consistent [113]. Crude oil is categorized into different groups based on API rating, including light, medium, heavy, and ultra-heavy. Guzman-Lucero et al. [113] investigated the demulsification efficiency of different ionic liquids for medium, heavy, and ultra-heavy crude oils, and they observed that all of the ionic liquids effectively destabilized emulsions of medium crude oil, but their efficiency was reduced for the ultra-heavy crude oil. High amount of natural emulsifying agents in the ultra-heavy crude oil led to the production of highly stable emulsions and the increment of oil viscosity, which would reduce the diffusion of ionic liquids. Ionic liquids with imidazolium, pyridinium, and ammonium cations with long alkyl chain length would be suitable for demulsifying heavy crude oils [113]. Similar results were obtained from the application of trioctylmethyl ammonium ethyl sulfate for demulsifying heavy and ultra-heavy crude oils, and a lower demulsification efficiency (50%) was observed for ultra-heavy crude oil than for heavy crude oil (70%) when 1500 ppm of trioctylmethyl ammonium ethyl sulfate was applied [102]. However, no significant changes were observed when TOMAC was used for medium, heavy, and ultra-heavy crude oils, and around 95% demulsification efficiency was achieved for all types of crude oils at an ionic liquid concentration of 1500 ppm and a treatment time of 6 h [102].

## 5. Challenges and Opportunities

In this review, the application of ionic liquids in demulsification processes and the influential factors are discussed. Along with the advantages of ionic liquids, there are still some challenges and opportunities regarding their demulsification applications.

### 5.1. Toxicity of Ionic Liquids

Generally, the toxicity of ionic liquids is some order of magnitude lower than conventional solvents such as acetone and methanol [125]. However, not all of the ionic liquids are environmentally friendly [126]. Toxicity of ionic liquids mainly depends on the cation type (e.g., imidazolium) and the cation alkyl chain length (e.g., >10 carbon atoms) [125,127]. Romero et al. [128] examined the toxicity of imidazolium-based ionic liquids with different alkyl chain length (1–8 carbon atoms), and they observed that toxicity of ionic liquids increased by increasing the alkyl chain length while anion type had less effect on the toxicity of ionic liquids [128]. It is reported that some of the ionic liquids containing fluoride and/or chloride ([BF_4_]-, [PF_6_]-) might generate hydrofluoric acid and/or hydrochloric acid in the presence of water [18,54,55]. The hydrolysis stability of anion should be high to prevent the formation of hydrofluoric acid and hydrochloric acid. Quijano [129] conducted a research to investigate the amount of the remained fluoride anion in the aqueous phase and evaluate its toxicity to microorganisms, and they applied C_4_mim PF_6_ and C_4_mim NTf_2_ ionic liquids at the concentration of 5 and 25% (volume/volume) in the mineral salt aqueous phase at the pH of 7 and the temperature of 25 °C. Their results indicated that fluoride anion in the aqueous phase was at very low concentration (0.73 to 2.98 ppm) which did not change the pH of mineral salt aqueous phase and was not toxic for microorganisms, and they also claimed that any changes in experimental conditions such as lower pH and higher temperatures (more than 25 °C) might have effects on the toxicity of fluoride anion [129]. Consequently, the toxicity of ionic liquids should be taken into consideration in their application for demulsification [54].

Bio-based ionic liquids (e.g., fatty acid ionic liquids) are considered as biodegradable ionic liquids with less or without significant toxicity [130,131]. They are made of natural-derived compounds, and their physical properties can be changed by manipulating their anion alkyl chain length. Longer anion alkyl chain length of fatty acid ionic liquids results in lower viscosity, density, water solubility (C_18_-stearate ionic liquid are not soluble in water), and corrosiveness as well as higher thermal stability [132,133,134]. Investigations indicated that the presence of hydroxyl group in the imidazolium cation part of fatty acid ionic liquids increased their thermal stability [132]. Biodegradability and toxicity of fatty acid ionic liquids are proportional to their physical properties (e.g., kinematic viscosity and water solubility) with linear relationships. Increased kinematic viscosity and water solubility would increase the biodegradability of fatty acid ionic liquids. Also, the toxicity of fatty acid ionic liquids decreased slightly by increasing the kinematic viscosity [135]. Also, fatty acid ionic liquids are non-fluoride-based hydrophobic ionic liquids, which can prevent the formation of acids in aqueous solutions.

Fatty acid ionic liquids were used successfully in various industries for different purposes, such as pharmaceutical [136], biorefining processes [137], solvents and co-solvents [138,139,140], dissolution and levulination of cellulose [141], formation of aqueous biphasic systems [142], CO_2_ capture [143,144], catalyst [145], lubricant additives [146], plant protection products [147], and extraction agents [148]. Therefore, the demulsification potential of bio-based ionic liquids, as well as their human health effects, are worthy of further investigations in future.

### 5.2. Viscosity of Ionic Liquids

Ionic liquids often have high viscosity which is undesired for their diffusion in a continuous phase. Viscosity of ionic liquids depends on the types of cation and anion. Selecting some types of anions (e.g., dicyanamide or bis(trifluoromethylsulfonyl)imide) in synthesizing ionic liquids can significantly reduce their viscosity [149]. However, these anions may not be effective for a variety of applications. Ionic liquids can be used with solvents (e.g., ethanol, methanol, and xylene) to reduce their viscosity. A research was conducted by Li et al. [150] to evaluate the effect of different solvents (e.g., chloroform, and acetonitrile, dichloromethane) on the properties of ionic liquids (C_4_mim PF_6_, C_4_mim BF_4_, C_4_mim CF_3_CO_2_) such as viscosity, and the results indicated that mixing ionic liquids in solvents noticeably reduced their viscosity. For example, the viscosity of C_4_mim PF_6_ reduced by 64% (from 261 mPa s to 92.9 mPa s) with addition of 0.2 mole fraction of acetonitrile [150]. Different solvents were reported to be applied with ionic liquids in many demulsification processes to increase the solubility of ionic liquids in oil phase. Xylene, methanol, 2-propanol, and ethanol are some of the solvents that were used to facilitate the dissolution of ionic liquids in oil phase [49,107,116,117]. It was reported that solvents with high dielectric constant (e.g., acetone, acetonitrile) can have a better effect in reducing the viscosity of ionic liquids than solvents with low dielectric constant (e.g., chloroform). Solvents with high dielectric constant have good miscibility with ionic liquids which can effectively reduce the electrostatic attraction between ions of ionic liquids and thus lowering the viscosity [150]. Therefore, the combination of ionic liquids with different solvents may be required prior to chemical demulsification to reduce the viscosity of ionic liquids. However, this might increase the costs of the entire demulsification treatment.

### 5.3. Recovery of Ionic Liquids

Although many commercial ionic liquids are available at a low cost, some of the best performers are still expensive for large-scale application compared to other commercial demulsifiers. Recovery and reuse of ionic liquids without compromising the demulsification efficiency becomes essential for their field-scale application because this can greatly reduce the cost [55,126]. There are many different methods to recover ionic liquids, such as liquid-liquid extraction, distillation, adsorption, crystallization, force field separation, and membrane processes [55,126,151]. The purity of ionic liquids is another important factor which should be considered in recycling or synthesizing ionic liquids. On one hand, impurities may lead to the production of unintentional by-products in the demulsification system; on the other hand, the impurities can change the expected characteristics of ionic liquids [54]. Among the aforementioned recovery methods, adsorption by activated carbons (ACs) is one of the common methods to separate ionic liquids from water streams. Lemus et al. [152] evaluated the efficiency of ACs with different structures in removing and recovering several imidazolium-based ionic liquids (e.g., C_8_mim PF_6_) from aqueous solution at the temperature of 34.85 °C, and their results indicated that most of the applying commercial ACs could separate at least 340 mg_C8mim PF6_/g_ACs_ from aqueous solution. The exhausted ACs could be regenerated using acetone extraction because the volatility and solvent capacity of acetone is high, and C_8_mim PF_6_ was recovered from regenerating acetone by atmospheric distillation at the temperature of 59.85 °C as characterized by H NMR spectroscopy. Based on their H NMR results, there was no difference in properties of the fresh and the recovered C_8_mim PF_6_, which indicated the successful recovery of C_8_mim PF_6_ [152].

### 5.4. Combination of Ionic Liquids with Nanoparticles

Recent investigations demonstrated that the presence of nanoparticles with ionic liquids or other demulsifiers in a system could enhance the interfacial properties and the demulsification process. The enhancement may be because demulsifier molecules adsorb on the surface of nanoparticles and create large particles which push them to move towards the O-W interface and break the interfacial film [153,154]. Another method was reported to coat ionic liquids onto the surface of nanoparticles to achieve promising results [155,156]. Atta et al. [155] in their research investigated the effect of 1-allyl-3-methylimidazolium oleate (AMO) coated magnetic nanoparticles (Fe_3_O_4_) at different concentrations (magnetic to oil ratio 1:10, 1:20, 1:25, 1:50) to remove oil from water, and found that 90% of oil was removed from water at the lowest concentration of magnetic nanoparticles capped with AMO (1:50). The magnetic nanoparticles were recycled for five times with a less reduction in their efficiency (efficiency reduced from 90% to 80% in the fifth cycle), and they concluded that magnetic nanoparticles can remove oil from water selectively without collecting water. These magnetic nanoparticles are easy to synthesize, cheap and reusable which could be applied in industrial scales [155]. Therefore, more efforts are needed to investigate the combinational demulsification effect of nanoparticles and ionic liquids to develop their applications in field-scales.

### 5.5. Poly Ionic Liquids

Synthesizing poly ionic liquids (PILs) has recently attracted research attention in oil field-related practices (e.g., EOR). PILs are polyelectrolytes consisting of polymeric backbone and an ionic liquid. In comparison with ionic liquids, PILs possess high activity at low concentrations as well as they are stable at high salinity and temperature [55,117,122]. Ezzat et al. [117] used ionic liquids and PILs based on 1,3-dialkylimidazolium to evaluate their efficiency on demulsification of W/O emulsions at different concentrations (50, 100, 250 ppm) and water contents (10%, 20%, 30%) at the temperature of 60 °C. Based on their results, PILs demulsified W/O emulsions better than their monomeric ionic liquids under the same experimental conditions (e.g., demulsification efficiency was 90% versus 70% at the concentration of 50 ppm and water content of 30% for EPDIB and EDDI, respectively) [117]. However, more investigations of PILs for demulsification under harsh environmental conditions are desired.

## 6. Conclusions

This review summarizes the recent advances in the application of ionic liquids as a chemical demulsifier for oil and water separation. Ionic liquids are promising demulsifiers, especially for applying under harsh environmental conditions characterized by high salinity and temperature as well as high viscosity (e.g., ultra-heavy crude oils). The main characteristics of ionic liquids that have attracted researchers’ attention are their thermal stability, non-flammability, recyclability, low vapor pressure, and low toxicity. Factors affecting their demulsification efficiency include ionic liquids types and concentrations, molecular weight, salinity, temperature, and types of oil in emulsions. The demulsification efficiency would be enhanced by selecting appropriate ionic liquids and dosage for specific types of emulsions as well as identifying optimal treatment conditions. Along with the advantages of ionic liquids, there are still some limitations which require further investigations to make them suitable for a wide application.

## Figures and Tables

**Figure 1 molecules-25-04915-f001:**
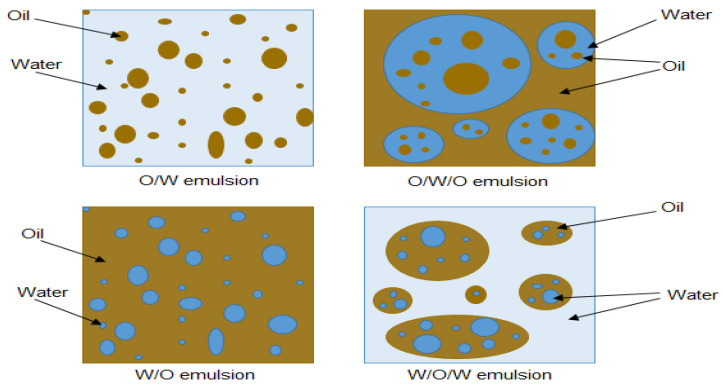
Different types of emulsions [25].

**Figure 2 molecules-25-04915-f002:**
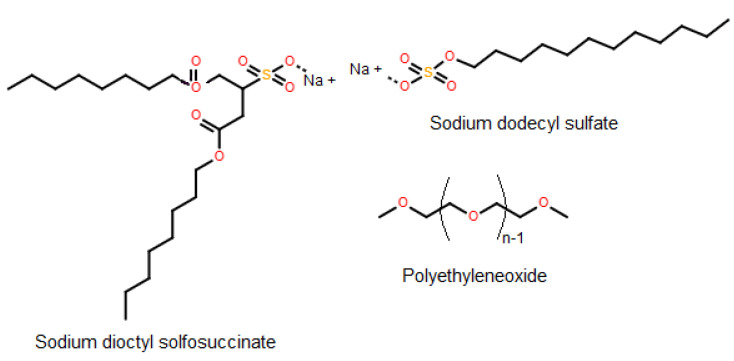
Structure of some common demulsifiers.

**Figure 3 molecules-25-04915-f003:**
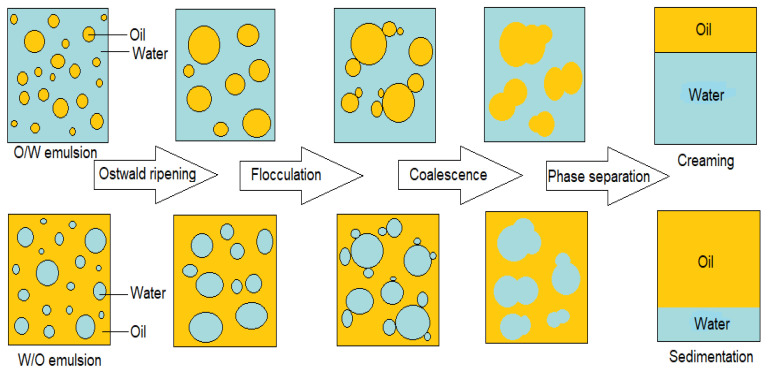
Schematic of chemical demulsification mechanism [31].

**Figure 4 molecules-25-04915-f004:**
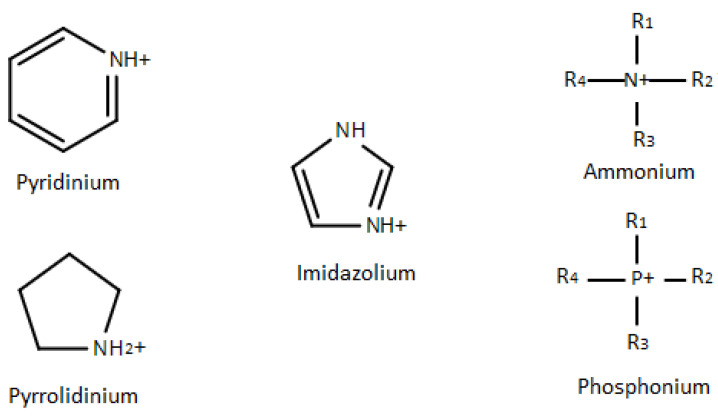
Common cations used for synthesizing ionic liquids.

**Figure 5 molecules-25-04915-f005:**
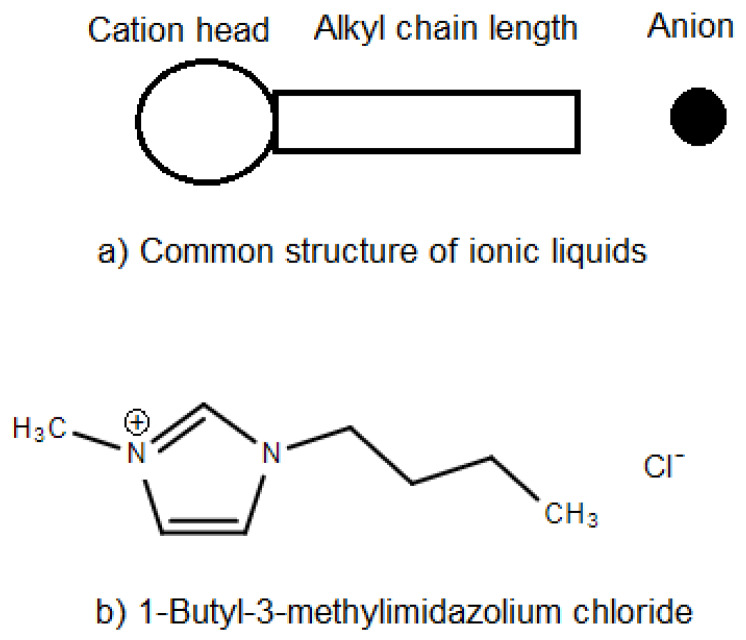
Structure of ionic liquids, (**a**) common structure, (**b**) structure of 1-butyl-3-methylimidazolium chloride [91].

**Table 1 molecules-25-04915-t001:** Melting point of common ionic liquids [85].

Ionic Liquid	Melting Point (°C)
C_2_mim BF_4_	15
C_2_mim TfO	−10.15
C_6_mim PF_6_	−61
C_8_mim BF_4_	−80
N_6222_ NTf_2_	20

**Table 2 molecules-25-04915-t002:** Different types of anions existing in ionic liquids [89,90].

Anion	Abbreviation	Types (Organic/Inorganic)
Alkyl sulfate	R-O-SO_3_^−^	Organic
Methane sulfonate	R_3_C-S-O_3_^−^	Organic
Tosylate	C_7_H_7_O_3_S^−^	Organic
Trifluoroacetate	CF_3_CO_2_^−^	Organic
Chloride	Cl^−^	Inorganic
Fluoride	F^−^	Inorganic
Bromide	Br^−^	Inorganic
Iodide	I^−^	Inorganic
Tetrachloroaluminate	AlCl_4_^−^	Inorganic
Hexafluorophosphate	PF_6_^−^	Inorganic
Tetrafluoroborate	BF_4_^−^	Inorganic
Bis(trifluoromethylsulfonyl) imide	[(CF_3_SO_2_)_2_N]^−^	Inorganic

**Table 3 molecules-25-04915-t003:** Guide for solvent selection for synthesizing as well as facilitating the dissolution of ionic liquids in oil phase [87].

Preferred	Usable	Undesirable
Acetone	Cyclohexane	Pentane
Ethyl acetate	Heptane	Hexane(s)
Water	Toluene	Di isopropyl ether
Ethanol	Methyl cyclohexane	Diethyl ether
Methanol	Isooctane	Dichloromethane
2-propanol	Acetonitrile	Dichloromethane
1-propanol	2-Methyltetrahydrofuran	Chloroform
Isopropylacetate	Tetrahydrofuran	Pyridine
1-butanol	Xylenes	Dioxane
Tert-butyl alcohol	Dimethyl sulfoxide	Dimethoxyethane
	Acetic acid	Benzene
	Ethylene glycol	Carbon tetrachloride
	Methyl Ethyl Ketone	

**Table 4 molecules-25-04915-t004:** List of application of ionic liquids as demulsifier.

Ionic Liquid	Cation Type	Anion Type	Emulsion Type	Dose (ppm)	CMC (ppm)	DE (%)	IFT Reduction (%)	Key Findings	Ref.
C_n_mim NTf_2_ n = 10, 12, 14	Imidazolium	bis(trifluoromethylsulfonyl)imide	SW/O	100–3500	N.A.	93.6–100	77–95	Demulsification experiments were conducted at the temperature of 60 °C. Increasing the dose and alkyl cation chain of hydrophobic ionic liquids improved the demulsification process (100% demulsification) as well as reduced the IFT (95% reduction). Higher hydrophobicity of NTf_2_ results in improving demulsification efficiency even with shorter alkyl chain length (e.g., demulsification efficiency was in the range of 93.6–100%). By contrast, for hydrophilic ionic liquid, increasing the dose and alkyl cation chain length of ionic liquid led to aggregation and caused poor demulsification as well as increased IFT.	[101]
C_n_mim PF_6_ n = 10, 12, 14		Hexafluoro phosphate		500–3500		71.25–86.25	54–81		
C_n_mim Cl n = 10, 12, 14		Chloride		500–3500		76.25–93.75	64–80		
TOMAC	Ammonium	Chloride	W/O	1000–2000	N.A.	100	N.A.	The efficiency of three ionic liquids (TOMAC, TOMAB, CTAB) with different hydrophobicity and hydrophilicity were evaluated for demulsification of W/O emulsions. Response surface methodology was applied to investigate the effect of temperature (50 °C–80 °C), pH (5–9), and water of aqueous phase (3–10%) on the demulsification efficiency. They observed that increasing ionic liquids concentration to 1039.2 ppm, 1480 ppm, 332.09 ppm for TOMAC, TOMAB, and CTAB, respectively led to the maximum demulsification efficiency (100, 90.29, and 64.9% for TOMAC, TOMAB, and CTAB, respectively). Demulsification efficiency increased at the pH of 7 and the temperature of 80 °C. Increasing water of emulsion (up to 10%) increased the demulsification efficiency of system to 64.88% and 90.29% using hydrophilic TOMAB and CTAB ionic liquids, respectively. Among three ionic liquids, TOMAC had the highest efficiency (100%) because it was more hydrophobic than other ionic liquids.	[107]
TOMAB		Bromide		1000–2000	N.A.	64.9			
CTAB		Bromide		300–700	N.A.	90.29			
Trihexyltetra decylphosphonium [Y]	Phosphonium	Chloride	W/O	50–4000	N.A.	71.42–99	N.A.	Experiments were conducted at different temperatures (60 °C and 80 °C) to investigate the efficiency of phosphonium based ionic liquids at different concentration of 50 to 4000 ppm on demulsifying W/O emulsions. Different hydrophobicity of anions led to different demulsification efficiency (e.g., varying from 14.29 to 99%).	[49]
		Decanoate				14.29–92.86			
		Dicyanamide				50–99			
AMPS/AA-TE	Oxyethylene ammonium	Sulfonate and carboxylate	W/O	100–500	0.00027 *	8–100	N.A.	Experiments were conducted at the temperature of 65 °C and different water content of emulsions (10, 20, 30, 50%). AMPS/AA-TE poly ionic liquid has oxyethylene in its structure which increased the polarity of AMPS/AA-TE. This led to AMPS/AA-TE having lower CMC than AMPS/AA-OA.	[36]
AMPS/AA-OA	Ammonium				0.00053 *	10–100			
TOMAC	Ammonium	Chloride	W/O	1500	N.A.	90	N.A.	Experiments were conducted at the temperature of 80 °C. TOMAC removed water from an extra heavy crude oil in less than an hour while two hours were required for trioctylmethyl ammonium ethyl sulfate and trioctylmethyl ammonium methyl sulfate to remove the same amount of water.	[102]
Trioctyl methyl ammonium [Y]		Ethyl sulfate					N.A.		
		Methyl sulfate					N.A.		
C_12_ mim NTf_2_	Imidazolium	bis(trifluoromethylsulfonyl)imide	W/O	5–125	100	N.A.	33.3	Applying ionic liquids with long alkyl chain length (12 carbon atoms) was more capable to displace the natural emulsifying agents of the crude oil which resulted in enhancing the IFT reduction (33%). Increasing ionic liquid concentration to CMC (100 ppm) reduced IFT, while no significant change was observed with concentration more than CMC.	[20]
HEOD-TS	Ammonium	Tosylate	SW/O	100–500	N.A.	30–100	95–99.5	Demulsification experiments were conducted at 65 °C. Increasing the concentration of hydrophobic HEOD-TS ionic liquid (e.g., from 100 to 500 ppm) for demulsifying SW/O emulsions at different water contents (10, 30, 50%) resulted in demulsifying emulsions completely (100%) as well as decreasing the IFT.	[24]
TOMAC	Ammonium	Chloride	W/O	1000 and 1500	N.A.	100	N.A.	Demulsification experiments were conducted at 80 °C using a water bath to remove water of two extra-heavy crude oils (with the water content of 56 and 60%). Increasing the concentration of ionic liquids from 1000 to 1500 ppm resulted in 100% demulsification efficiency. Ionic liquids with smaller anion size have lower anion polarizability which result in dehydration of extra-heavy crude oils effectively.	[108]
Trioctylmethylammonium [Y]		Bisulfate							
		Dihydrogenphosphate							
C_8_mim PF_6_	Imidazolium	Hexafluorophosphate	W/O	600–6200	N.A.	54.7–95.6	92	High dosage of ionic liquids resulted in 95.6% and 87.4% of demulsifying W/O emulsions using C_8_mim PF_6_ and C_8_mim BF_4_, respectively that were implemented under microwave heating (90 °C) and different water content of emulsions (~ 30 to 50%). C_8_mim PF_6_ decreased the IFT and separated water from oil more effective than C_8_mim BF_4_ (95.6 and 87.4% for C_8_mim PF_6_ and C_8_mim BF_4_, respectively). The reason is that C_8_mim PF_6_ has bigger anion size and lower solubility in water which prevents the aggregation of ionic liquid in the medium in comparison with C_8_mim BF_4_.	[114]
C_8_mim BF_4_		Tetrafluoroborate		1000–7200		0–87.4	85		
C_4_mim NTf_2_	Imidazolium	bis(trifluoromethylsulfonyl)imide	SW/O	0.74–8.9 **	N.A.	10	1	Imidazolium and pyridinium based ionic liquids were used to demulsify W/O emulsion (water content of 40 wt%) at the temperature of 120 °C. There was no significant difference in the demulsification efficiency between the imidazolium and the pyridinium ionic liquids with the same alkyl chain length and anion type. Ionic liquids with longer alkyl chain (e.g., 8 and 12 carbon atoms) were more capable to displace the natural emulsifying agents of the crude oil which resulted in enhancing the demulsification process (74% and 90% for C_8_mim NTf_2_ and C_12_mim NTf_2_, respectively) as well as IFT reduction. Higher hydrophobicity of NTf_2_ results in improving demulsification efficiency (e.g., 74% and 40% for C_8_mim NTf_2_ and C_8_mim OTf, respectively).	[115]
C_8_mim NTf_2_						74	4		
C_12_mim NTf_2_						90	34		
C_8_mim OTf		triflate				~40	N.A.		
C_4_py NTf_2_	Pyridinium	bis(trifluoromethylsulfonyl)imide				10>	N.A.		
EDHI	Imidazolium	Acetate	W/O	50–250	N.A.	0–70	N.A.	Demulsification process using imidazolium-based ionic liquids were conducted at 60 °C and different water contents (10, 20, 30%). Increasing ionic liquids concentration from 50 to 250 ppm increased the demulsification efficiency to 70, 85, 100, and 100 for EDHI, EPHIB, EDDI, and EDPIB, respectively at different experimental conditions. Using 4-(trifluoromethoxy)phenylborate anion increased the hydrophobicity of EPHIB compared to EDHI which resulted in enhancing the demulsification process (e.g., demulsification efficiency increased from 70 to 85%). Based on the results, the efficiency of polymeric ionic liquids was better than that of their monomeric ionic liquids.	[117]
EPHIB		4-(trifluoromethoxy) phenylborate				10–85			
EDDI		Acetate				70–100			
EPDIB		4-(trifluoromethoxy) phenylborate				85–100			
P_666,14_ (CN)_2_	Phosphonium	Dicyanamide	O/W	***	N.A.	100	N.A.	In this research, different ionic liquids with hydrophobic cation and hydrophilic anions were used to demulsify O/W emulsions at room temperature. P_666,14_[N(CN)_2_] had high surface active area and removed oil from water completely. However, stable emulsions still existed in the systems that P_666,14_[Phos], P_666,14_[NTf_2_] and N_2224_[N(CN)_2_] were used because surface active area would not achieve in too hydrophobic (P_666,14_[Phos], P_666,14_[NTf_2_]) and too hydrophilic (N_2224_[N(CN)_2_]) ionic liquids.Halogenide ionic liquids, P_666,14_[Cl] and P_666,14_[Br] separated oil from water in a very short time (20 min) compared to non-halogenide ionic liquid P_666,14_[N(CN)_2_] (24 h).	[118]
P_666,14_ Phos		bis(2,4,4-trimethylpentyl) phosphinate				0			
P_666,14_ NTf_2_		bis(trifluoromethylsulfonyl)imide				0			
P_666,14_ Cl		Chloride				>90			
P_666,14_ Br		Bromide							
N_2224_ N(CN)_2_	Ammonium	Dicyanamide				0			
C_8_mim Cl	Imidazolium	Chloride	W/O	100–10000	1000	N.A.	3–73	In this research, imidazolium and pyridinium based ionic liquids were used at different concentration (100–1000 ppm) and different temperatures (20 °C to 60 °C) to evaluate their efficiency on IFT reduction. Pyridinium cation is more hydrophobic than imidazolium cation, therefore pyridinium based ionic liquids can remain at the O-W interface better than imidazolium based ionic liquids, which enable them to reduce the IFT of crude oil-distilled water effectively at lower CMC.	[99]
C_12_mim Cl					2000		6–84		
C_8_Py Cl	Pyridinium				N.A.		4–45		
C_12_Py Cl					500		7–93		

Note: DE: Demulsification efficiency. SW: Saline water. N.A.: Not available. [Y]: Refer to anion type. * Mole/Liter. ** μmole/gram of emulsion. *** Mole ratio of ionic liquid:sodium dodecylbenzenesulfonate is 1.

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
