# Peer review of "Application of Ionic Liquids for Chemical Demulsification: A Review"

_molecules, 2020, doi:10.3390/molecules25214915_

Round 1
Reviewer 1 Report
The review deals with the literature survey about the demulsification potential and limits of ionic liquids. The review introduces the topic also for non-experts in the field rendering the paper attractive to a broader audience. The review is well organized and the effects of structural changes of ionic liquids and of variations of operational parameters on the demulsification performances are analyzed. The topic is of interest and further investigation in the field, which could shed light on the pro and cons of ionic liquids use, are needed and expected.
However, two sections should be improved.
- General literature about the properties and use of ionic liquids should be implemented with additional more relevant works.
Please add the following papers to the highlighted sections:
Pharmaceutical: 10.1021/acssuschemeng.0c03501
Oil & gas: 10.1021/acssuschemeng.9b06336
Chemical Industries: 10.1021/acssuschemeng.0c03951
Pharmaceutical manufacturing: 10.1039/c9cc02812a, 10.1016/j.molliq.2020.112547, 10.3390/pharmaceutics12100909
Liquid-Liquid extraction: 10.1016/j.molliq.2019.111841, 10.1021/acssuschemeng.0c00429
Polymeric Ionic liquids: 10.1021/acssuschemeng.0c00327, 10.3390/polym12091909
Dicationic Ionic liquids: 10.1016/j.jcou.2019.07.034, 10.1016/j.molliq.2020.112983, 10.1021/acssuschemeng.0c02473
Deep eutectic solvents: 10.1002/cssc.202001331, 10.1039/D0GC00793E, 10.1002/cssc.201900147.
Solvate ionic liquids: 10.1039/C4CP00461B
Line 176: add to ref 73 the following works: 10.1021/ie5009597, 10.1039/C3CS60071H
2) Toxicity. As this is a very important and hot topic in the ionic liquid area, it is strongly advised to add potential alternatives for the future developments of ionic liquids for demulsification studies, with a special emphasis on fatty acid ionic liquids which are one of the few options of non-fluoride-based hydrophobic ionic liquids. Therefore the section should mention the following works.
Biobased ILs: 10.1039/c6gc01482h, 10.1039/d0gc01002b, 10.1021/acssuschemeng.6b00553, 10.1039/C9NJ00191C, 10.1016/j.fluid.2019.05.001, 10.1002/ejic.202000364, 10.1016/j.cogsc.2016.09.001, 10.3390/nano9040504, 10.1016/j.molliq.2020.112607 .
Fatty acids ILs: 10.1016/j.molliq.2019.111155, j.molliq.2020.112827, 10.1016/j.molliq.2019.111444, 10.1080/01932691.2018.1541416, 10.3390/molecules24050894
Toxicity Fatty acids Ionic Liquids: 10.1016/j.molliq.2019.111451, 10.1021/acssuschemeng.8b06201, 10.1016/j.molliq.2019.111822, 10.1039/d0ra00766h, 10.1016/j.foodres.2020.109125.
Minor changes requested:
Line 159 and line 165: please replace “buthyl” with “butyl”
Ref 53 please insert Volume and pages.
Therefore, I recommend publication of the proposed manuscript after the above requested changes.
Reviewer 2 Report
This manuscript provides a detailed review of the application of ionic liquids for chemical demulsification. The manuscript was well-written with the scope and purpose of the review clearly indicated. The basic information was provided for a broader readership and the structure was well-organized covering all the important aspects. The topic of the manuscript is not of great scientific importance but of enough application interest. I would recommend publishing the manuscript after a minor revision, taking the following question into account:
The effect of cations and anions were discussed separately for the specific properties. How does the interaction of cation and anion affect the results? Will adding certain anion change properties of cations hence affecting the overall property? Will cation and anion compete with each other for the oil/water interface? Please comment.
Round 2
Reviewer 1 Report
The Authors improved the manuscript in accordance to my suggestions. Therefore, I recommend the publication of this work in the present form.